# The Link between VAPB Loss of Function and Amyotrophic Lateral Sclerosis

**DOI:** 10.3390/cells10081865

**Published:** 2021-07-23

**Authors:** Nica Borgese, Nicola Iacomino, Sara Francesca Colombo, Francesca Navone

**Affiliations:** CNR Institute of Neuroscience, Via Follereau 3, Bldg U28, 20854 Vedano al Lambro, Italy; nicola.iacomino@istituto-besta.it (N.I.); sara.colombo@in.cnr.it (S.F.C.)

**Keywords:** endoplasmic reticulum, FFAT motif, membrane contact sites, motor neurons, neurodegeneration, phosphoinositides, VAP proteins

## Abstract

The VAP proteins are integral adaptor proteins of the endoplasmic reticulum (ER) membrane that recruit a myriad of interacting partners to the ER surface. Through these interactions, the VAPs mediate a large number of processes, notably the generation of membrane contact sites between the ER and essentially all other cellular membranes. In 2004, it was discovered that a mutation (p.P56S) in the *VAPB* paralogue causes a rare form of dominantly inherited familial amyotrophic lateral sclerosis (ALS8). The mutant protein is aggregation-prone, non-functional and unstable, and its expression from a single allele appears to be insufficient to support toxic gain-of-function effects within motor neurons. Instead, loss-of-function of the single wild-type allele is required for pathological effects, and *VAPB* haploinsufficiency may be the main driver of the disease. In this article, we review the studies on the effects of VAPB deficit in cellular and animal models. Several basic cell physiological processes are affected by downregulation or complete depletion of VAPB, impinging on phosphoinositide homeostasis, Ca^2+^ signalling, ion transport, neurite extension, and ER stress. In the future, the distinction between the roles of the two VAP paralogues (A and B), as well as studies on motor neurons generated from induced pluripotent stem cells (iPSC) of ALS8 patients will further elucidate the pathogenic basis of p.P56S familial ALS, as well as of other more common forms of the disease.

## 1. Introduction

Amyotrophic lateral sclerosis (ALS) is an adult-onset incurable disease, with an incidence of about 1/50,000 per year, defined by the degeneration of both upper and lower motor neurons (MN). MN degeneration is followed by muscle denervation and atrophy; death, due to respiratory failure, usually occurs within three years from the initial diagnosis. Most cases of ALS arise in individuals without a family history of the disease and are thus referred to as sporadic ALS (sALS); in contrast, ~10% of cases, termed familial ALS (fALS), are transmitted within families [1].

Although fALS is far less common than sALS, much effort has been devoted to identifying its genetic causes and to characterising the underlying cellular pathways. Indeed, because of the similarity of the clinical pictures of the sporadic and familial forms, and because a genetic component is implicated also in sALS [2,3,4], it is thought that unravelling the mechanisms of fALS pathogenesis will eventually also translate to a better understanding of sALS.

Since the discovery of the first fALS-linked gene—the one coding for Cu/Zn superoxide dismutase (*SOD1*- [5])—nearly 30 monogenic disease genes plus additional genetic risk factors have been discovered [4,6]. Currently, the identified loci account for approximately 70% and 10% of fALS and sALS cases, respectively [6]; thus, at nearly thirty years from the recognition of the causal link between SOD1 and fALS, the search for additional genes is still on.

The numerous disease genes discovered so far have been broadly classified into a limited number of functional categories, comprising RNA metabolism, DNA damage response, mitochondrial functionality, protein homeostasis (proteostasis), protein trafficking, cytoskeletal and axonal dynamics [7,8,9]. Even so, the numerosity and diversity of the disease genes suggest multiple mechanisms of ALS pathogenesis: the disease is likely to be triggered by perturbations of different processes that all converge on to final, ALS-defining, MN death programmes. An in depth understanding of the different events upstream to the death pathways may lead to the recognition of different sALS subtypes, with prospects for the development of targeted therapeutic strategies.

In 2004, a missense mutation in the *VAPB* gene (p.P56S) was discovered to be the cause of a dominantly inherited slowly progressing form of fALS (known as ALS8), as well as of typical rapidly progressing fALS and spinomuscular atrophy, in a large cohort of Brazilian families of Portuguese origin [10]. VAPB and its paralogue VAPA are integral membrane proteins of the endoplasmic reticulum (ER), which, because of their prominent role in tethering the ER to the cytosolic surface of all other organelles, are key players in interorganellar communication ([11,12]—see Section 2).

The mutation in all the affected Brazilian families was inherited from a single founder living in the mid-15th century in Portugal, from where the mutation was brought to Brazil during the colonial period [13]. Subsequently, the same *VAPB* p.P56S mutation was discovered in unrelated subjects in North America [14], Germany [15], and China [16]. Other fALS-linked *VAPB* mutations have since been discovered [17,18,19] but have been less investigated than p.P56S.

From the time of its identification as disease gene, the role of *VAPB* in ALS pathogenesis has been intensively investigated, in parallel with explosive developments in the understanding of the multiple roles of the VAP proteins in cellular physiology (reviews: [20,21,22]). These developments have provided a unique basis to unravel at the molecular level the link between VAPB and ALS pathogenesis, as will be discussed in this review article. The interest is heightened by the observation that VAP levels are decreased in sALS patients’ cells, as well as in a mouse model of fALS caused by a gene unrelated to VAPB [23,24,25,26]. Hence, although *d* mutations are rare, understanding the role of the encoded protein in fALS8 is expected to yield insights into the more common sporadic forms of the disease.

## 2. The VAP Proteins: Structure and Function

The VAMPs-associated proteins (VAPs), so named because of their capacity to associate with vesicle associated membrane protein (VAMP) [27], are tail-anchored proteins of the endoplasmic reticulum, which function as adaptors, recruiting a large variety of proteins to the cytosolic surface of the ER (reviews: [20,21,22]). At the N-terminus, the VAPs present an immunoglobulin-like seven-stranded β sandwich known as the major sperm protein (MSP) domain [28,29,30], which is crucial to VAP functions; this is followed by a predicted coiled-coil region, and, at the C-terminus, a hydrophobic sequence, which anchors the VAPs to the ER bilayer (Figure 1A). Vertebrates express two VAP paralogues, VAPA and B, which share a large part of their sequence, especially in the MSP domain (82% identity). The VAP proteins can form homodimers as well as VAPA-B dimers, thanks to interactions between their transmembrane (TM) helices and predicted coiled-coil regions [31,32,33].

The VAPs function by recruiting protein ligands from the cytosol to the surface of the ER, but can also associate with some protein partners within the ER bilayer, via interactions between the TM domains (Figure 1A). The association of many (but not all) of the cytosolic partners is mediated by the so-called FFAT motif (two phenylalanines in an acidic tract—consensus sequence: EFFDAXE) within the ligand [36], which binds to an electropositive region running transversely across four of the seven β strands of the MSP, flanked by two hydrophobic pockets [28,29,30]. Studies aimed at defining the full complement of VAP interactors have revealed that significant deviations from the initially defined consensus are tolerated [21,37,38,39]. The complexity of the VAP interactome (VAPome) is further increased by the discovery that some motifs that deviate from the initially defined consensus are regulated by phosphorylation (phospho-FFAT motifs [40,41,42]). The uniquely large number of interactions of the VAP proteins underlies the very many functions that they carry out, as summarised in Table 1.

An important subset of VAP interactors serve as bridges that connect the ER to essentially every other organelle in the cell, leading to the close apposition (~5–30 nm) of regions of the ER and tethered organelle membranes, to form structures known as membrane contact sites (MCS). These structures, whose importance has been recognised relatively recently, allow communication between organelles in the absence of fusion between the bounding membranes, and are, notably, sites of lipid and Ca^2+^ exchange [11,12,43,44].

In summarising VAP functions (Table 1), we have made no distinction between the two VAP paralogues, because the degree of overlap between VAPA and B functions is as yet unclear. Indeed, systematic investigations of the VAP interactome have failed to reveal any important differences between the two paralogues (e.g., [38]). Furthermore, to our knowledge, there is no one function that has been clearly demonstrated to be carried out exclusively, or preferentially, by VAPB. Elucidation of the division of labour between the two VAPs would represent an important step towards understanding the pathogenic mechanism of VAPB mutations. To be noted, invertebrates have only one VAP orthologue, which presumably carries out all the essential functions of the two vertebrate VAPs.

We highlight here three additional points, which are relevant to the understanding of the link of the *VAPB* gene with fALS8:(1)In most analysed tissues, including spinal cord, as well as in cell lines, *VAPA* is expressed at higher levels than *VAPB*, both at the mRNA and at the protein level (www.genecards.org; www.ebi.ac.uk/gxa; www.gtexportal.org [38]). For instance, in a murine model MN cell line (NSC34) VAPA was found to be ~5 fold more abundant than the B paralogue [94]), and an even higher excess was reported for HeLa cells [95].(2)In mice, *VAPA* knockout is embryonic lethal [91], whereas deletion of VAPB is compatible with survival into adulthood [96]. The difference in the tolerance of the animals to knockout of each of the paralogues could be due to the higher abundance of VAPA, so that its absence results in a larger reduction of the total VAP pool than does VAPB deletion; alternatively, or in addition, the different sensitivity could be due to the existence of an essential function of VAPA that VAPB cannot carry out.(3)Given the very many functions that the VAPs are involved in (Table 1), it is impossible to assign the *VAPB* gene a priori to any of the functional categories in which ALS-linked genes have been classified (Section 1). Indeed, it fits into many of them.

## 3. The p.P56S Mutation: Loss or Gain of Function?

The proline residue, which is substituted with serine as a consequence of the p.P56S mutation, is in a conserved position of the MSP domain of VAPB (see Section 2), adjacent to the FFAT binding site. The substitution of Pro with Ser at this position causes the protein to become aggregation-prone [30,33,97], and indeed large P56S-VAPB aggregates are readily observed in transfected cells [10,23,32,98,99,100,101] and in tissues of transgenic animals [102,103,104,105,106,107,108,109,110]. Aggregation results in a profound restructuring of the portion of the ER where the mutant protein is concentrated, as revealed by ultrastructural analyses [100,101]. Indeed, the P56S-VAPB inclusions consist in stacks of a small number of undulating ER cisternae (generally two or three) separated by an electron-dense layer of cytosol. Within these inclusions, the mutant protein is unable to interact, or interacts poorly, with FFAT-containing ligands [23,33,99,105,111].

Despite the dramatic phenotype induced by overexpressed P56S-VAPB, the inclusions it forms are easily cleared by the proteasome [101,112] and thus are unlikely to accumulate significantly in patients’ cells, in which the mutant protein is expressed from a single allele. In agreement, P56S-VAPB inclusions have not so far been detected in MNs generated from ALS8 patients’ induced pluripotent stem cells (iPSC) [113] or in patients’ cultured fibroblasts [14].

Because of the dominant inheritance of ALS8, and because of the role that protein aggregates play in many neurodegenerative diseases, including ALS [114], many studies following the discovery of the p.P56S mutation addressed the effects of the mutant protein overexpressed in cells and in animals. Depending on the system investigated, the aggregates were observed to sequester an ion channel, preventing its transport to the cell surface [76], and to trap proteins involved, for instance, in membrane traffic [73], in proteostasis [104,105,106,109], in the generation of MCS [60,69]—but see also [111]), in phosphoinositide metabolism [110]. These observations were consistent with the idea that the P56S-VAPB inclusions could be harmful to MNs by a toxic gain-of-function mechanism. In addition, and importantly, many studies found that the inclusions can sequester the product of the wild-type *VAPB* allele, both in transfected cells [23,32,33,99] and in transgenic animals [107,108,109], supporting a dominant negative loss-of-function pathogenic mechanism of the mutant. The significance of all these studies is, however, limited by the high levels of overexpression of the mutant protein, not consistent with those obtained when expression is from a single allele, as in ALS8 patients’ cells. Contrasting the hypothesis of a pathogenic action of P56S-VAPB aggregates, a critical survey of the animal studies leads to the conclusion that the expression of the mutant protein is, by itself, insufficient to cause MN disease, as discussed in more detail in a recent review article [115]. The main arguments supporting this conclusion are summarised in the next paragraphs.

Of the four transgenic mouse lines that have been generated [102,103,104,105], only one, in which the mutant protein was highly overexpressed, developed motor symptoms [104], despite the presence of VAPB aggregates in the MNs in all four of them. Similarly, in *Drosophila,* a model organism that has been extensively used to model ALS8 [116,117], most studies have been carried out under conditions of overexpression; when moderate and similar levels of the mutant protein and wild-type protein were co-expressed, no pathology nor reduced survival were observed [118]. In stark contrast, a knock-in mouse, in which the mutant gene replaces the wild-type one, did develop motor symptoms and partial denervation of lower MNs [106]. Importantly, such symptoms developed also in heterozygote mice, which, similar to ALS8 patients, bear one copy of the wild-type and one copy of the mutant allele. These results, in comparison with those obtained with the transgenic models described above, argue strongly in favour of the idea that loss of one functional *VAPB* allele is necessary for the development of MN disease, by a mechanism of haploinsufficiency. It remains to be demonstrated, however, that haploinsufficiency alone is sufficient for the development of the full-blown disease. Analysis of *VAPB*-knockout mice (generated and analysed in a laboratory different from the one that generated the knock-in model) revealed very mild motor impairment in the homozygote and none in the heterozygote [96], however, a direct comparison between the knockout and knock-in mouse lines has not been carried out.

Despite the absence of observable P56S-VAPB aggregates in patients’ iPSC-derived MNs and fibroblasts [14,113], it is possible that some aggregates do accumulate in ageing MNs, and that the dominant negative effect or toxic gain-of-function of these aggravates the situation caused by VAPB haploinsufficiency. To be noted, although it was initially reported that VAPB inclusions sequester VAPA in primary hippocampal neurons [23], this sequestration was weak in comparison with the one of VAPB, and others have failed to confirm this observation [99,103,111]. Thus, the aggregates, if present in ALS8 patients’ MNs could be interfering with wld-type VAPB function, but VAPA is predicted to be undisturbed.

From the above discussion, we argue that the understanding of the mechanistic basis of the pathogenicity of the p.P56S mutation involves determining which of the many VAP functions is lost upon the exclusive depletion of VAPB, in the face of unaltered VAPA levels. The remaining part of this review will discuss studies that have been carried out with just that aim, and will focus on four functions reported to be impaired as a consequence of VAPB deletion, all of which have important implications for neurodegeneration (Figure 1B,C). We will not, instead, discuss the studies involving the overexpression of mutant *VAPB*, nor those reporting the effects of combined depletion of the two VAP paralogues in mammals or the depletion of the unique invertebrate orthologue. Whilst these studies have yielded important insights into VAP function in general, they are less directly relevant to the pathogenic effects of *VAPB* mutation in MNs. We also will not be discussing the important work on non-cell-autonomous effects in ALS pathogenesis, including the extracellular functions of a secreted form of VAP [109].

## 4. Effects of VAPB Depletion in Cellular and Animal Models

### 4.1. ER-Mitochondria Contacts (Figure 1C-Box 1)

Although sites of contact between the ER and mitochondria in fixed tissue were observed in early EM studies (e.g., [119] and references therein), the first functional role of these contacts was demonstrated by Vance in 1990 [120]. In this work, a subcellular fraction of mitochondria-associated ER membranes (MAMs) was shown to be enriched in a subclass of ER-localised lipid synthesising enzymes, consistent with their role in supporting a collaborative effort of the two compartments (ER and mitochondria) in phospholipid homeostasis. Indeed, most lipids are synthesised in the ER and from there must be delivered to other organelles. Within the exo-endocytic pathway, this can be in part effected by transport vesicles, however, because mitochondria do not participate in vesicular traffic but do depend on the ER for most of their lipids [121], other transport mechanisms must be at play. The work of Vance opened the way for further study of the ER-mitochondria contact sites where this lipid transport occurs [122].

Following the studies on lipid transfer, a second key role of ER-mitochondria contacts, the facilitation of mitochondrial Ca^2+^ uptake, was demonstrated [123]. Because of the low affinity for Ca^2+^ of the mitochondrial calcium uniporter (MCU) [124], the average cytosolic [Ca^2+^] reached even after its stimulated release from ER stores is a poor driver of transport of this cation into mitochondria [125]. However, Ca^2+^ release at contact sites exposes mitochondria to a higher [Ca^2+^], which matches the MCU’s low affinity and allows for rapid Ca^2+^ transfer from the ER to the mitochondrial matrix. Subsequent studies indeed demonstrated that the inositol-3-phosphate responsive Ca^2+^ release channel of the ER (IP3 receptor) and the outer mitochondrial membrane channel, which permits the passage of Ca^2+^ ions into the intermembrane space (VDAC, voltage-dependent anion selective channel), are physically coupled via the cytosolic chaperone protein grp75 [126]. The rapid uptake of Ca^2+^ by mitochondria from ER stores is crucial for the stimulation of rate-limiting enzymes of the Krebs cycle and consequent ATP production in response to agonists [127], as well as for the rapid buffering of cytosolic Ca^2+^.

The elucidation of the roles of ER-mitochondria contacts in lipid transport and Ca^2+^ regulation was followed by the discovery of additional functions of these contacts [128], and the identification of proteins that link the two organelles. In yeast, an elegant genetic screen led to the discovery of the ERMES complex (ER-mitochondrial encounter structure) [129]. This structure is not present in higher eukaryotes, but intensive research has uncovered many other tethering complexes, which may have specific roles in the different processes governed by ER-mitochondria contacts (reviews: [11,128]). Of the numerous complexes identified so far, four contain the VAPs linked to different FFAT-containing proteins on the mitochondrial surface, specifically: protein tyrosine phosphatase-interacting protein 51 (PTPIP51) [60], vacuolar protein sorting (VPS) 13A/ D [42,59,130], and mitoguardin 2 (MIGA2) [65,66] (see Table 1).

The interaction of VAPB with outer mitochondrial membrane protein PTPIP51 [also known as FAM82A2, human cerebral protein-10 and regulator of microtubule dynamics protein-3 (RMD-3)] was discovered in a two-hybrid screen designed to retrieve VAPB partners [60] and confirmed in subsequent proteomic screens [38,131]. The molecular basis of its interaction with VAP has not been fully elucidated so far: a FFAT-like motif is present in the central part of the sequence [39,111], and mutation of VAPB’s FFAT binding site abrogates the interaction [38], however the VAPB MSP by itself is not sufficient to pull down PTPIP51, suggesting that VAP regions outside of the FFAT binding site contribute to the binding [61].

Quite remarkably, silencing of either VAPB or PTPIP51 in a model motorneuronal cell line (NSC34) led to a ~40% reduction in the extension of ER-mitochondria contacts [61]. This effect may be cell-specific, as it was not observed in HeLa cells [95]. Considering the multiplicity of proteins that act as tethers between the two organelles [11,128], a ~40% reduction is an impressive effect. Depletion of VAPB or PTPIP51 had similar effects, and PTPIP51 silencing reduced the amount of VAPB recovered in a mitochondria plus MAM subcellular fraction, suggesting that PTPIP51 is VAPB’s major partner on the outer mitochondrial membrane of MNs [60,61].

The basis for the large effect of VAPB depletion on ER-mitochondria contacts is currently incompletely understood, given that PTPIP51 interacts also with VAPA [38,131], as well as with three other ER-localised MSP domain-containing proteins (the motile sperm protein domain (MOSPD)-containing proteins [38,95]). In NSC34 cells, VAPA is present in 5-fold excess over VAPB [94], indicating that, despite its higher abundance and its ability to interact with PTPIP51, it cannot compensate for the lack of VAPB. It is possible that PTPIP51 has a higher affinity for VAPB than for the A paralogue, as suggested by co-immunoprecipitation experiments [38]. Another possibility is that the tethering process is mediated by VAP heterocomplexes, of which VAPB is an essential component.

An important consequence of the loosening of ER-mitochondria contacts caused by VAPB depletion is the delayed and reduced uptake of Ca^2+^ into mitochondria after stimulated release of the cation from ER stores [60]. In agreement with the role of the VAPB-PTPIP51 tether in mitochondrial Ca^2+^ uptake, overexpression of either of the two proteins increases the IP3 Receptor-VDAC interaction [62].

Stimulated by the findings on VAPB-PTPIP51-mediated ER-mitochondria contacts and on the role of Ca^2+^ in mitochondrial energy metabolism [127,132], we recently investigated whether the chronic depletion of VAPB in MN-like NSC34 cells affects mitochondrial function. In published work, we have extensively characterised this cell line, and reported alterations in phosphoinositide homeostasis and neurite elongation [94], as described in the following subsection of this review. As illustrated in Figure 2, we observed a decrease in uptake of the mitochondrial membrane potential sensor tetramethylrhodamine methyl ester perchlorate (TMRM) into the mitochondria of the VAPB-depleted cells, suggestive of reduced oxidative phosphorylation. This observation is in agreement with the reported reduced ATP production by oxidative phosphorylation in neuronal cell lines or primary cortical neurons under conditions in which ER-mitochondria contacts are loosened [64,133]. The relevance of these findings to ALS is strengthened by computational models that predict that even small decreases in ATP availability may disrupt neuronal ion homeostasis and functionality [134]. Importantly, VAPB-PTPIP51-mediated contacts are present in nerve terminals, and VAPB or PTPIP51 silencing in primary hippocampal neuron cultures reduces ER-mitochondria contacts concomitantly with synaptic activity [135].

Another observed consequence of VAPB or PTPIP51 depletion is an increase in autophagic flux. Conversely, overexpression of either of the two proteins decreased autophagosome formation [62]. The effect of VAPB and PTPIP51 depletion on autophagic flux was directly ascribed to the loosening of ER-mitochondria contacts and to the consequent decreased ER to mitochondria Ca^2+^ transfer: indeed, expression of an artificial tether rescued the autophagic phenotype, while interference with Ca^2+^ release from the ER or uptake into mitochondria abrogated the inhibitory effect of the tether [62]. These findings have implications for the pathogenic mechanism underlying ALS8, as perturbed regulation of autophagy is a common feature of neurodegenerative diseases [136,137].

Beyond ALS8, VAPB-mediated ER-mitochondria contact sites may be relevant to ALS in general, as suggested by the effect of two ALS-linked gene products, Tar-DNA binding protein 43 (TDP43 or TARDBP) and fused in sarcoma (FUS), on the VAPB-PTPIP51 tether. Both are DNA and RNA-binding proteins, involved in DNA repair and RNA transcription and processing. Although mainly localised to the nucleus of healthy cells, they are often found in pathological cytosolic aggregates in sALS, fALS, and frontotemporal dementia patients’ cells [138,139]. TDP43 aggregates, in particular, which contain ubiquitinated, hyperphosphorylated, and cleaved forms of the protein, are present in a majority of ALS cases [139] and considered a hallmark of the disease. In addition, mutations of both *TARDPBP* and of *FUS* are linked to a subset of fALS cases [140,141,142], providing a link between sALS and fALS pathogenesis. Very interestingly, expression of wild-type or disease-associated mutants of *TARDBP* and *FUS* cause a decrease in VAPB-PTPIP51 interaction and a concomitant decrease in the extension of ER-mitochondria contacts in NSC34 cells and transgenic mice, in parallel with perturbation in Ca^2+^ handling [61,133]. In both cases, activation of glycogen synthase kinase 3β was linked to the disruption of the VAPB-PTPIP51 tether.

Disruption of ER-mitochondria contacts as a common theme in neurodegenerative disease, including ALS, is attracting increasing interest and could represent a point of convergence of pathological events triggered by different initial insults to neurons [34,64,143,144,145,146]. This fascinating subject is discussed in greater detail in another review of this special issue [147].

### 4.2. Regulation of Phosphatidylinositol-4-Phosphate (PI4P) (Figure 1C-Box 2)

PI4P is one of the two most abundant phosphoinositides, a family of signalling lipids that are generated by phosphorylation of the inositol headgroup of phosphatidylinositol (PI) and that act by recruiting specific proteins to the cytosolic surface of membranes. Because of the restricted distribution of the kinases, which generate them, the phosphatases, which consume them, and lipid transport proteins (LTPs) which transport them, each phosphoinositide is specifically distributed to single compartments or subsets thereof, thereby contributing to the definition of organelle identity (reviewed in [148]). PI4P, in particular, is considered the signature phosphoinositide of the Golgi complex, but is also present in lysosomes/late endosomes (LE) and at the plasma membrane [149]; in these locations, it is involved in a myriad of essential processes, including lipid homeostasis, membrane trafficking, autophagy, signalling at the plasma membrane, and actin dynamics (reviews: [150,151]).

In the Golgi complex, PI4P levels critically depend on the VAPs and on the contact sites between the ER and the *trans*-Golgi that they mediate. Initially visualised by EM cytochemistry [152] and then by high resolution tomography [153], these contacts have been recently characterised at the molecular level with a novel Förster energy transfer (FRET)-based technique [154]. This approach has identified the VAPs in conjunction with members of the oxysterol binding protein (OSBP)–related protein (ORP) family (see below) as required for contact formation. Since ORPs are LTPs, the VAP-ORP connection has the dual role of establishing a physical connection between the ER and the *trans*-Golgi, and mediating lipid transport at these sites.

LTPs are proteins that effect non-vesicular lipid transport between membranes, thanks to a hydrophobic cavity within their structure that allows them to shield the transported lipid from the surrounding aqueous environment [155]. There are many known LTP classes; of these, OSBP and related ORPs are unique in their ability to transport PI4P and to exchange this phosphoinositide for another lipid [156]. This ability is due to the lipid-binding OSBP-related domain (ORD) in the C-terminal part of the polypeptide, which can accommodate either one PI4P or another lipid molecule (cholesterol or phosphatidylserine) [157].

In addition to the ORD, OSBP and all ORPs, except two, contain, in the middle of their sequence, an FFAT motif for VAP binding and, towards the N-terminus, a PI4P-specific pleckstrin homology (PH) domain (with the exception of ORP2 and alternatively spliced forms of some other ORPs). The PH domain mediates ORP binding to the membrane on the non-ER side of the contact site, a binding which usually occurs by coincidence detection of PI4P in conjunction with a membrane-associated protein, e.g., the small GTPase Arf1 in the case of the OSBP-Golgi connection. Thus, ORPs use PI4P for membrane recognition by the PH domain and transport this phosphoinositide via the ORD (reviewed in [157]).

The role of the founding member of the ORP family, OSBP, in cholesterol and PI4P transport at ER-Golgi contact sites has been investigated in detail. The low concentration of PI4P in the ER favours binding of cholesterol (produced in the ER) to OSBP’s ORD. On the Golgi side, the higher concentration of PI4P allows binding of the phosphoinositide in exchange for cholesterol; back in the ER, PI4P is released, and the cycle is repeated. The low concentration of PI4P in the ER is maintained by the ER resident phosphoinositide 4-phosphatase SAC1, which removes the 4-phosphate group from PI4P to generate PI [activity in cis [158]]. PI is then returned to the Golgi by PI transport proteins (Nir2, see Table 1) and serves as substrate for PI4 kinases to regenerate PI4P. Thus, the transport of cholesterol out of the ER against its concentration gradient is fuelled by the ATP required to regenerate PI4P and to maintain the Golgi-ER gradient of this phosphoinositide [45,157,159].

In the OSBP cycle, the VAPs have a dual function: first, they provide the anchoring site for OSBP, second, VAPA modulates SAC1 activity. In *Drosophila* [110] and in mammals [49,74], a direct interaction between VAP and SAC1 has been demonstrated. SAC1 lacks an FFAT motif, and differing conclusions on the protein region(s) involved in the interaction have been reported: in *Drosophila*, they were mapped to the TM domains of both SAC1 and VAPA [110]; in mammalian cells, instead, Venditti et al. [49] found that an N-terminal, cytosol-exposed fragment of SAC1 is sufficient for the interaction, and that both the catalytic and the TM domain are dispensable. The VAP-SAC1 interaction presumably enhances SAC1 efficacy by concentrating the enzyme at ER-Golgi contacts, where it encounters high concentrations of PI4P back-transported to the ER.

Under certain conditions, SAC1 can act on the Golgi pool of PI4P, thereby decreasing the Golgi/ER concentration gradient. Indeed, under starvation, SAC1 is transported to the Golgi, where it reduces PI4P content in situ [160,161]. In addition, the recruitment of SAC1 to particular contact sites, established by the FFAT containing protein FAPP1, allows the phosphatase to act in *trans* on Golgi PI4P at sites of high concentration of the phosphoinositide [49]. Thus, SAC1 both maintains the PI4P Golgi/ER gradient and regulates the Golgi PI4P pool.

Given the above-described roles of the VAPs in the OSBP cycle, it is not surprising that the combined depletion of VAPA and B, or of the single VAP orthologue of invertebrates, causes an increase in PI4P levels [49,58,110,162] (a caveat is that the transport of PI to the Golgi, too, is mediated by the VAP-dependent LTP Nir2! [50]). Less expected, yet more relevant to ALS8 pathology, is the observation that also the exclusive depletion of VAPB, with VAPA levels untouched, has this effect [77,94], indicating a critical role of VAPB in phosphoinositide homeostasis. Interestingly, in the MN-like NSC34 cell line, even a partial reduction of VAPB, to levels mimicking those of a p.P56S-*VAPB* heterozygote, was sufficient to affect PI4P levels in the Golgi complex and to cause the expansion of a PI4P-positive population of peripheral vesicles. Some, but not all of these vesicles were found to be acidic and to co-localise with the LE/lysosome marker LAMP1 [94]; no other markers of the endocytic pathway or of the *Trans*-Golgi Network (TGN), such as Rab7 or the mannose-6-phosphate receptor, were found within them (our unpublished observations; see also subsection on delayed neuritogenesis, below).

An open question is the identity of the VAPB-sensitive step(s) within the complex cycle of events that regulate PI4P intracellular levels. Indeed, OSBP associates with both VAP paralogues [21,38,131], and, in pulldown experiments, SAC1 has a strong preference for VAPA over the B isoform [49,74]. As stated earlier in this review (Section 2), there is more VAPA than B in most cells, including MNs, so one might anticipate that VAPB is dispensable for processes that function equally well with either paralogue. Notably, however, silencing of VAPB in HeLa cells was reported to alter the localisation of transfected GFP-SAC1 [77]. In addition, other ORPs which might preferentially associate with the VAPB paralogue, could be involved in generating the phosphoinositide imbalance [49,77].

Regardless of the mechanism by which VAPB depletion causes the PI4P increase, the question that we address in the following part of this section concerns the effects of this increase on cellular physiology. In *Drosophila* and *C. elegans* the consequences of PI4P elevation are disastrous, as depletion of SAC1 or of the single invertebrate VAP orthologue causes alterations of synaptic morphology and neurodegeneration, a phenomenon linked to PI4P: indeed, reduction of the phosphoinositide by downregulation or pharmacological inhibition of PI4 kinases rescues both neurodegeneration and the synaptic phenotype of these model organisms [110,163]. In mammals, downregulation of VAPB alone, in the presence of functional VAPA, is expected to have less dramatic effects, however, two PI4P-linked phenomena observed in VAPB-depleted mammalian cells—the maintenance of nuclear envelope architecture in HeLa cells and neurite elongation in NCS34 cells—may contribute to MN degeneration in ALS8.

#### 4.2.1. Nuclear Envelope Defects

In HeLa cells, *VAPB* silencing causes delocalisation of the inner nuclear membrane protein emerin from its normal localisation at the nuclear ring to cytoplasmic puncta [77,164]. Emerin, whose loss of function causes Emery-Dreifuss muscular dystrophy, is a LAP2-emerin-MAN1 (LEM) domain protein [165] that plays a key role in the organisation of the nuclear lamina and its association with chromatin, as well as in transcriptional regulation, mitosis, and nuclear assembly (reviewed in [166]). The important roles of emerin in establishing and maintaining nuclear envelope organisation likely explain why several nuclear pore proteins were observed to be delocalised in *VAPB*-silenced cells [164].

In the study of Darbyson and Ngsee [77], emerin delocalisation occurred also after silencing of *ORP3* or *SAC1*, and in all three cases (*VAPB*, *SAC1* or *ORP3* silencing), it was paralleled by augmented PI4P; the effect of VAPB silencing was partly reversed by overexpression of *ORP3*, suggesting that the VAPB-emerin link is mediated by PI4P imbalance. How this imbalance could affect emerin targeting is currently unclear. Possible lipid compositional changes, which could occur consequent to the PI4P imbalance and affect targeting of nuclear envelope proteins, have not been investigated so far. A further twist to the story is given by the results of a recent study, which demonstrated that VAPB itself localises to the inner nuclear membrane, where it directly interacts with emerin, and also with the nuclear pore protein ELYS [78]. These results suggest that VAPB may play a direct role in emerin targeting and nuclear pore assembly.

The nucleus-VAPB link has important implications, because defects in the nuclear envelope, delocalisation of nuclear pore proteins, and impaired nucleo-cytoplasmic transport, have all been linked to ALS pathogenesis [167,168], and defective nucleo-cytoplasmic transport has been observed in cultured fibroblasts of an ALS8 patient [14]. Further analysis of ALS8 patients’ cells will hopefully confirm a role of emerin delocalisaton in triggering MN degeneration.

#### 4.2.2. Delayed Neuritogenesis

Genevini et al. generated NSC34 clones with different degrees of VAPB downregulation. When these VAPB-depleted cells were induced to differentiate, a significantly reduced rate of neurite elongation was observed, suggesting a defect in the trafficking of transport vesicles to the growing neurite. The delay was observed in both the near completely silenced and in the partially downregulated cells (which also had increased PI4P levels), suggesting that in wild-type cells the concentrations of VAPB are near threshold for normal function ([94], see Figure 3). A causal relationship between delayed neuritogenesis and elevated PI4P was indicated by the rescue of the phenotype when the levels of the phosphoinositide were reduced by pharmacological inhibition of PI4 kinase IIIβ, which is active at both the Golgi complex ([169] and lysosomes [170,171]). Although the study of Genevini et al. analysed differentiating cells, the observed neuritogenesis defect could be relevant for the survival of mature MNs, as these must continuously replenish their axons and dendrites with membrane components. Beyond ALS8, disrupted neuronal trafficking is thought to be an important player in the pathogenesis of ALS types linked to other, more common, disease-causing genes [172].

The molecular mechanism(s) underlying the inhibitory effect of excess PI4P on neuritogenesis have not been deciphered yet. PI4P, by recruiting adaptors and regulators at the TGN, has a crucial role in the generation of secretory vesicles [150,173], and augmented PI4P at the TGN can increase secretion of some substrates [49]. The effects of altered PI4P levels at the Golgi are, however, probably cell- and substrate specific: for instance, deletion of SAC1 in mammalian cells inhibits transport of a substrate out of the TGN in constitutive carriers [74]. A block in transport could also occur at a stage subsequent to carrier budding at the TGN. Analogously to the situation described in yeast, where post-Golgi vesicles must loose PI4P to become competent for polarised exocytosis [174,175], an excess of PI4P on post-Golgi vesicles of differentiating NSC34 cells could reduce the vesicles’ capacity to fuse with the plasma membrane. The PI4P-containing acidic vesicles that we observed in the VAPB-depleted NSC34 cells could represent stalled intermediates on their way to fusion with the plasma membrane. Alternatively, they could represent a population of hypofunctional lysosomes, reported to accumulate in *Drosophila* cells as a consequence of elevated PI4P levels caused by the deletion of the *Drosophila* VAP homologue [162]. Whether the PI4P build-up in these vesicles is due to the PI4P overload in the TGN, or is a direct consequence of VAPB deficit-caused impairment of ER–endolysosome contact sites, has not been established so far.

While the PI4P excess caused by VAPB depletion may be acting directly on vesicular traffic, for instance by aberrant recruitment of PI4P-binding proteins, it probably interferes with membrane trafficking by indirect effects too: indeed, alteration of the quantity and distribution of lipids that depend on ORP-PI4P driven countertransport and of the related regulatory feedback loops could affect the generation of membrane carriers at the TGN and exocytosis [176,177,178].

In summary, VAPB deficit leads to excess PI4P by mechanisms that have not yet been entirely clarified; the PI4P excess causes neurodegeneration in model organisms, probably by interfering with more than one essential process. Defects in nuclear envelope maintenance and neurite elongation are two readouts of PI4P excess that are likely important contributors to the final death of MNs.

### 4.3. Hyperpolarsation-Activated Cyclic Nucleotide-Gated (HCN) Channels 1 and 2 (Figure 1C-Box 3)

By allowing the entrance of a depolarising mixed Na^+^ and K^+^ current (the I_h_ or I_f_ current) in response to hyperpolarisation, HCN channels, expressed mainly in the heart and in the CNS, play a key role in controlling the rhythmic activity of cardiac pacemaker cells and spontaneously firing neurons. Mammals produce four HCN paralogues, each containing six TM helices, which form channels by homo- or hetero-tetramerisation. Gating of the channels in response to the membrane potential is modulated by cyclic AMP, which binds to a site close to the C-terminus of each channel subunit. Relevant to the subject of this review, in neurons, HCN channels are involved in several functions additional to rhythmicity regulation, among which determination of resting membrane potential and regulation of excitability, dendritic integration, and synaptic transmission (reviewed in [179]).

In a two-hybrid split-ubiquitin screen designed to identify proteins associated with the HCN2 channel, VAPB turned up in a large number of positive clones [76]. The interaction, confirmed by pulldown experiments, and extended to HCN1, but not HCN4, was mapped to VAPB’s TM helix and to the N-terminal portion of the HCN channel.

The functional outcome of the association was tested in microinjected *Xenopus* oocytes, and in transfected or VAPB depleted mammalian cells, as well as in animal models. In *Xenopus* oocytes and in HeLa cells, co-expression of VAPB with HCN2 increased I_h_ amplitude by favouring the surface expression of the channel without altering its electrophysiological properties. This effect appeared to be independent from recycling of the channel and was therefore attributed to its improved transport through the secretory pathway to the cell surface. Interestingly, co-expression of VAPB’s TM helix alone recapitulated the effects obtained with the full-length protein. In Zebrafish and in a knockout mouse [32], VAPB depletion caused severe bradycardia, and, in the knockout mouse, a reduction in the amplitude of I_h_ in neurons of the ventrobasal thalamus, consistent with reduced excitability. No data on HCN channel-mediated currents in MNs of the knockout mice were provided; however, both the HCN1 and 2 paralogues are expressed in MNs of the brainstem and of the spinal cord [180,181], suggesting that they play a role in these neurons, too. The results reported by Silbernagel et al. provide a potential and very interesting link between VAPB deficit and ALS8 pathogenesis, in light of the implication of ion channel dysfunction as a common theme in neurodegenerative diseases [182].

A challenging question left open by the work of Silbernagel et al. [76] is the mechanism by which VAPB facilitates the surface expression of HCN1 and 2. As noted above, the interaction is mediated by VAPB’s TM helix, and therefore occurs within the plane of the bilayer, a finding difficult to reconcile with the different localisations (ER and plasma membrane) of the interacting proteins. The authors suggest that VAPB travels through the secretory pathway together with the channel, with which it would remain associated as an accessory subunit; this hypothesis, however, awaits rigorous proof. As an alternative possibility, we propose that VAPB could transiently interact with the channel in the ER, thereby facilitating the assembly of the subunits into a tetramer competent for export to the cell surface. In other words, it would act as an intramembrane chaperone. Whether VAPB is a stable accessory channel subunit or a transiently interacting protein in the ER, this interesting and novel VAP function deserves further investigation.

As in the case of ER-mitochondria contacts and of PI4P homeostasis (Section 4.1 and Section 4.2), also for the VAPB-HCN channel interaction the question arises as to the cause of the important effect of VAPB in the face of normal VAPA levels. This case, however, may provide perhaps the first example of a clear-cut preference of a client for VAPB over the A paralogue. Indeed, most of the results of the study of Silbernagel et al. suggest that HCN channels interact preferentially with VAPB: in the two-hybrid screen, there were no VAPA hits, and VAPA was ineffective in pulling down transfected HCN channels; VAPA was also less effective than VAPB in increasing I_h_ current amplitude when co-expressed in Xenopus oocytes. Nevertheless, additional clarification is warranted, because the endogenous channel, as well as the in vitro translated polypeptide, interacted much more strongly with VAPA than VAPB in pulldown assays. Despite this caveat, it is tempting to speculate that differences in binding partners between the two VAP paralogues are more likely for interactions that involve the TM helices than those involving other protein domains; this is because of the poor conservation of the TM helices in comparison to other VAP regions (e.g., 39 and 82% sequence identity in the TM and MSP domains, respectively, between human VAPA and B). Further studies, involving site-directed mutagenesis within the TM domains, will shed light on the molecular basis of TM-based interactions of the VAP proteins.

### 4.4. Unfolded Protein Response (UPR), ER Stress and Protein Quality Control (PQC) (Figure 1C-Box 4)

#### 4.4.1. Adaptive and Maladaptive UPR

The UPR is a central cellular signalling hub that governs homeostasis, development, and life-death decisions, and whose deregulation has been implicated in a variety of diseases, including neurodegeneration (reviews: [183,184,185]). In mammals, the UPR is mediated by three transmembrane sensors, which detect problems in the ER lumen (unfolded proteins) and initiate signal transduction pathways aimed at resolving these problems: (i) the transmembrane kinase/RNAse IRE1 initiates the unusual cytosolic splicing of an mRNA that codes for the transcription factor X-box binding protein-1 (XBP1); XBP1s, generated from the spliced mRNA, upregulates the expression of genes for components of the ER folding machinery and of the ER-associated degradation (ERAD) system; in addition, the RNAse activity of IRE1 may cleave, hence inactivate, ER-associated mRNAs coding for other proteins by a process designated regulated IRE1-dependent decay (RIDD), thereby reducing the translational load on the ER; (ii) ATF6 is a transcription factor, which, under basal conditions is integrated in the ER membrane by a TM helix. In response to stress, ATF6 is transported by vesicular carriers to the Golgi apparatus, where two proteases act in sequence to release its active domain; this can then enter the nucleus and complement the transcriptional activity of XBP1s; (iii) the third sensor, PKR-like ER kinase (PERK), by phosphorylating the translation initiation factor eIF2α, attenuates general protein synthesis but allows the preferential translation of some transcripts, among which the one coding for the transcription factor ATF4.

If the cellular responses driven by the three UPR branches are successful, the end result is restored homeostasis. Indeed, the ER improves its capacity to deal with the overload, because, on the one hand, its protein folding capacity is upregulated, and, on the other, it more efficiently disposes of malfolded proteins by the ERAD pathway; at the same time, the overload is reduced because of attenuated protein synthesis. If, however, the beneficial interventions of the UPR are insufficient to compensate for the initial genetic or environmental problem that triggered the response, the ER becomes chronically stressed. As a consequence, the UPR, faced with a situation sensed as unresolvable, may switch from its attempts to restore homeostasis to an activity, designated as maladaptive, that favours apoptosis. Understanding the basis of the transition from the adaptive to the maladaptive UPR is currently one of the most challenging aspects of research on cellular homeostatic mechanisms.

Several apoptotic pathways are involved in ER stress-triggered cell death. First, the cytosolic region of IRE1 can interact directly with the adaptor TRAF2, thereby activating the stress-activated kinase Jun N-terminal kinase (JNK) [186]; second, ATF4 and its target gene product, CHOP (C/EBP homologous protein), are transcription factors that control genes involved in apoptosis [187]; third, the RIDD activity of IRE1 [188] can result in the degradation of mRNAs encoding proteins required for survival [189] as well as of miRNAs that regulate the expression of apoptotic proteins [190].

#### 4.4.2. The UPR at the Intersection of PQC Pathways

As mentioned above, the disposal of irreversibly unfolded/misfolded proteins in the ER lumen is handled by the ERAD system; this PQC pathway effects the retro-translocation (also referred to as dislocation) of the undesired substrates back to the cytosol and their delivery to the ubiquitin-proteasome system (UPS). The UPS, a key pathway for the regulated degradation of short-lived proteins, also provides a major mechanism for disposal of misfolded proteins and protein aggregates (reviewed in [191,192]). Because the disposal of cytosolic and ER lumenal proteins converge on this same system, it follows that any disturbance of it, even if originating in the cytosol, may have consequences for the ER too. For instance, pathological aggregates of the Huntington disease protein polyQ-Huntingtin, by clogging the UPS, result in impaired ERAD function, decreased misfolded protein flux, and ER stress [193,194]. Not to be forgotten are the effects of oxidative stress, which can be driven by pathological protein aggregates, on protein folding in the ER and on Ca^2+^ handling, again illustrating how events in the cytosol reverberate on the ER; this subject is discussed in more detail in another article of this special issue [195].

In addition to the intersection with the UPS, there is also crosstalk between ER stress and the other major system for degradation of macromolecules, the lysosome-based macroautophagy (autophagy) system. In this pathway, cellular material, such as protein aggregates or damaged organelles, are first sequestered within double-membrane bounded structures named autophagosomes and then degraded after fusion of the autophagosomes with lysosomes (reviewed in [196]). ER stress has generally been found to inhibit autophagy but may in some cases have the opposite effect [197]. Importantly, the UPS and autophagy exert reciprocal influence on each other [198]. Thus, the UPS, the UPR, ERAD, and autophagy are all part of the same PQC network, whose proper functioning is crucial for cellular health and survival.

#### 4.4.3. ER Stress and ALS8

As discussed above, a maladaptive UPR may be the outcome of different initial insults to the cell and deal a final blow to their survival. ER stress has indeed been implicated in a number of neurodegenerative diseases [199]; In ALS, a longitudinal study of a commonly used mouse model—the superoxide dismutase 1 (*SOD1*) transgenic mouse—revealed that the more vulnerable MNs are those that exhibit a UPR before showing signs of degeneration, suggesting a causal link between ER stress and MN death [200]. Because of the Jekyll-and-Hyde nature of the UPR, however, an adaptive UPR may instead help insulted MNs to withstand stress [201].

Turning to ALS8, the available, limited, information suggests that, like in typical ALS, ER stress may be involved in the development of the disease. Indeed, an analysis of a patient’s cultured fibroblasts reported a significant increase in the expression of genes associated with the PERK-driven apoptosis pathway, including *CHOP* and *ATF4*, as well as in *ATF6* and the spliced mRNA encoding XBP1s; some chaperones were also upregulated, however, no increase in the master UPR chaperone and luminal stress sensor binding protein (BiP, alias glucose regulated protein of 78 kDa (GRP78)) was detected [14]. In partial agreement with these results, the spinal cord of heterozygote p.P56S-*VAPB* knock-in mouse showed a significant increase in the phosphorylation of the translation initiation factor eIF2α, the immediate readout of PERK activation, which preceded morphological alterations of the neuromuscular junction and partial muscle denervation [106]. Although increased phosphorylation of eIF2α was not observed in the patients’ fibroblasts, and, conversely, *CHOP*, *ATF4* and *ATF6* upregulation was not observed in the p.P56S-*VAPB* knock-in mouse, the results suggest that the ATF4/CHOP apoptotic pathway may be activated in ALS8 MNs. It should, however, be kept in mind that eIF2α phoshorylation is the point of convergence of four different stress-responsive kinases, of which only PERK is a UPR transducer [202,203]. Thus, activation of this pathway, known as the integrated stress response, may be triggered by problems in the cytosol rather than directly by ER stress.

In addition to alterations in stress markers, the heterozygote p.P56S-*VAPB* knock-in mice showed increased levels of the ubiquitin receptor p62/SQSTM in spinal cord MNs [106]; p62/SQSTM is involved in the delivery of ubiquitinated protein aggregates both to the proteasome and to autophagic membranes (reviewed in [204]). This observation suggests a deregulation of PQC in MNs, which, the authors suggest, could underlie the observed ER stress [106]. Notably, p62/SQSTM mutations are linked to fALS and frontotemporal dementia and have been identified in sALS cases too [205,206].

The above-described results in patients’ cells and in the p.P56S-*VAPB* knock-in mouse lead to the question as to whether the observed ER stress and deregulated PQC are direct effects of VAPB deficit or the indirect consequence of the problems associated with its loss of function, as described in Section 4.1, Section 4.2 and Section 4.3. For example, loosening of ER-mitochondria contacts augments basal autophagy [62], an effect that could clash with the effects of increased PI4P concentration in the endosome-lysosome compartments. Indeed, as mentioned in Section 4.2, expansion of an endosomal pool deriving from a PI4P-overloaded Golgi apparatus was reported to result in hypofunctional lysosomes [162], and excess incorporation of the phosphoinositide in the autophagosomal membrane inhibited its fusion with lysosomes [207]. In addition, a contribution to stress of mutant *VAPB* by gain-of-function mechanisms is possible. As discussed in Section 3 of this review, P56S-VAPB aggregates have not been detected in patients’ cells, however, they may accumulate in ageing MNs and aggravate the stress caused by *VAPB* haploinsufficiency.

A possible direct role of VAPB in PQC regulation and ER stress has been investigated in acutely depleted cells and in protein-protein interaction studies. Two studies on the effects of *VAPB* silencing on the UPR in mammalian cultured cells generated different results, most likely because of the choice of different cells and reporters: in NSC34 cells, VAPB deficit reduced the UPR in cells exposed to ER stressors, suggesting that it has a positive role in triggering ER stress [32]; in HEK293 cells, instead, *VAPB* silencing caused an increase in basal, as well as tunicamycin-induced, ATF6/XBP1s-dependent transcription, suggesting that it plays a negative role in blunting the UPR [75]. A possible mechanism underlying this UPR attenuation is the observed interaction between the VAPs and ATF6. This interaction was reported to be mediated by the VAP MSP and an unidentified region of ATF6’s transcription factor domain [75,208], but has not, to our knowledge, been further characterised.

Another potentially very interesting VAPB interaction, involved in PQC, is the one with the p97 adaptor FAF1/ UBXN3A/UBXD12 [83]. The FAF1 adaptor belongs to a family of ubiquitin-binding adaptors for p97/Valosin Containing Protein 1 (VCP1), an AAA ATPase that, by extracting misfolded proteins from the ER for proteasomal degradation, is a central player in ERAD. The interaction with FAF1 is mediated by a non-canonical FFAT motif and involves an acidic loop in the VAP MSP additional to its canonical FFAT binding site [83,208]. FAF1 mediates recruitment of the VAPs to complexes involved in ERAD at the ER membrane [208] and their association with polyubiquitinated proteins [83]. Expression of VAPB and A in cultured mammalian cells was observed to stabilise a well-known ERAD substrate, and it was suggested that VAP’s interaction with ERAD components blunts their activity [208]. Notably, mutations in p97/VCP1 are linked to fALS, and have been identified in sALS cases too [209,210].

The VAPs also interact with a number of autophagy proteins, which play roles in autophagosome biogenesis and in receptor-mediated clearance of the ER (ERphagy), as summarised in Table 1. To our knowledge, however, the extent of the specific dependence of autophagy and ERAD-related phenomena on the VAPB paralogue has not been investigated so far.

In summary, the current results suggest that chronic ER stress and PQC dysregulation are present in ALS8 MNs, however, further studies are required to confirm and fully characterise the perturbed pathways. It will be particularly challenging to distinguish the direct consequences of loss-of-function of *VAPB* from stress responses due to upstream effects of its deficit, and to understand the possible role of the mutant gene’s product in perturbing protein homeostasis.

## 5. Conclusions and Outlook

Since the initial discovery of the VAPs [27], a large body of research has focused on the exceptionally numerous interactions and functions of these highly versatile proteins. Interest was heightened by the discovery, in 2004, of the link between *VAPB* and ALS; hence, progress in basic research on VAP cell/molecular biology over the past ~15 years has proceeded in parallel with work aimed at understanding ALS8 pathogenesis. Many studies have been directed at clarifying the pathogenic mechanisms of the ALS-linked p.P56S mutation; however, a review of all the studies carried out on animal and cellular models up till now leads to the conclusion that the mutant protein itself is insufficient to cause disease, and that a deficit of the wild type protein is the main driver of pathological changes in MNs (reviewed in [115]). Accordingly, *VAPB* loss-of-function has been shown to perturb several basic cell physiological processes, as discussed in this review.

Despite the progress in elucidating the link between *VAPB* loss-of-function and neuronal degeneration, there are still many open questions on the sequence of events leading to the death of ALS8 MNs. Given the pleiotropic effect of *VAPB* loss-of-function, it remains to be elucidated which of the many VAPB functions/binding partners are critical for MN health and survival. Is it the sum of more than one defective process that triggers disease, or does one of the affected processes play the major pathogenic role? Furthermore, since there appears to be more VAPA than VAPB in most tissues, including MNs, the detrimental effect of VAPB depletion on MNs points to a specific role of VAPB, not carried out by VAPA, or to a function that critically depends on the full VAP pool. We believe that an important step forward will be to address the problem of VAPB versus VAPA specificity: for instance, what are the relative roles of the two paralogues in PTPIP51-mediated ER-mitochondria contacts and in HCN channel transport to the cell surface? What is the target of VAPB in regulation of the PI4P pool? Are other partners specific to VAPB involved in MN degeneration? Is ER stress a direct consequence of VAPB depletion or secondary to other deficits caused by its insufficiency?

Important advances in the understanding of the *VAPB*-ALS link are likely to be obtained from the further characterisation of iPSC-derived MNs from ALS8 patients. VAPB aggregates have not been observed in these cells, but so far there are no data on their survival, their electrophysiological properties, their PI4P levels in the Golgi. More research on these cells could open the way to investigate the effects of drugs that target specific cellular pathways, with the prospect of developing novel therapeutic approaches; these could be relevant not only for ALS8 but also for more common typical ALS or for a subset of ALS cases that share a VAP deficit with ALS8. Indeed, as pointed out in the introductory section of this review, VAP deficit has been implicated in *SOD1*- linked and sporadic ALS, and malfolded VAPB in peripheral blood mononuclear cells has been suggested as a possible diagnostic biomarker for ALS [26]. Furthermore, VAP deficit in *Drosophila* glial cells has been proposed to cause neuroinflammation, suggesting a VAPB-linked non-cell autonomous mechanism underlying MN death [211]. These phenomena broaden the scope of future research aimed at generating knowledge on the VAPs as well as on the pathogenic mechanisms of *VAPB* loss-of-function, which will hopefully translate to the clinic.

## Figures and Tables

**Figure 1 cells-10-01865-f001:**
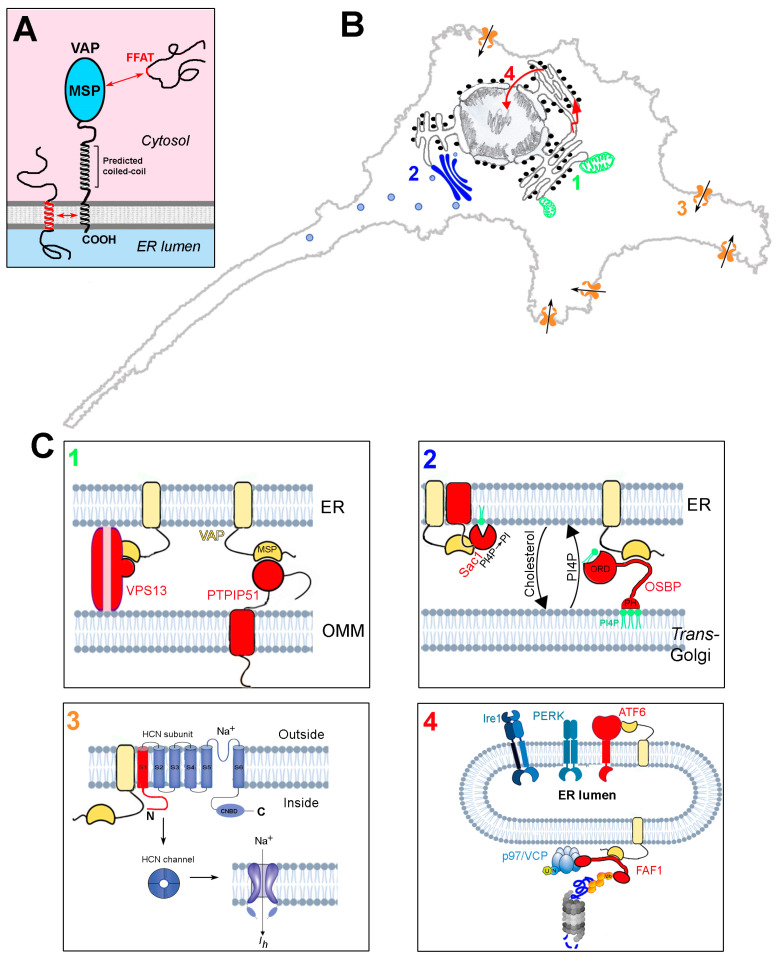
Structure and interactions of the VAP proteins. (**A**) Domain organisation and membrane topology of the VAPs. Interactions between FFAT motifs and the MSP domain, as well as those involving the TM helix, are indicated. (**B**) Schematic representation of four sites of action of the VAPs, involving processes that are perturbed by the deficit of the VAPB paralogue. Each of the four processes, and how they are affected by VAPB deficit, are discussed in Section 4.1, Section 4.2, Section 4.3 and Section 4.4. 1: ER-mitochondria contact sites; 2: ER-Golgi contact sites; 3: regulation of HCN channels; 4: UPR and PQC. (**C**) Zoom-up images of sites 1–4 of panel (**B**), illustrating the underlying VAP interactions. For each site, the interacting proteins (or portions thereof) are shown in red. VAP is in yellow. Interactions between the MSP and FFAT motifs are indicated by complementary surfaces, which are purposely not drawn in cases where an FFAT motif is not involved in the interaction. VAP and PTPIP51 are redrawn from Ref. [34] and the FAF1 interactions (panel 4) are adapted from [35] (Creative Commons license). Phospholipid bilayers, UPR sensors, and the proteasome are adapted from Biorender (https://biorender.com/).

**Figure 2 cells-10-01865-f002:**
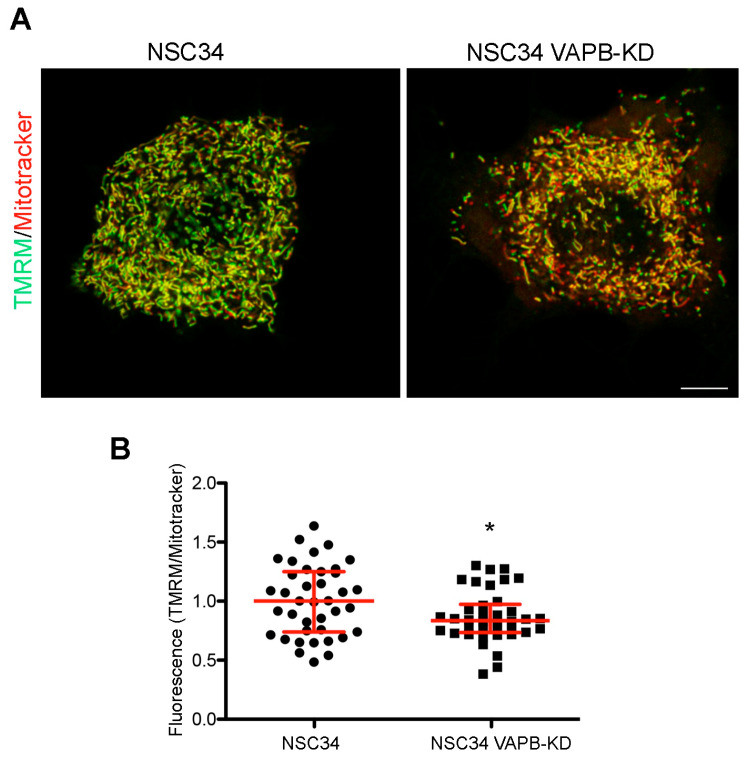
VAPB knockdown decrease mitochondrial membrane potential in the MN-like cell line NSC34. (**A**) Representative images comparing staining with the membrane potential-sensitive and insensitive mitochondrial dyes TMRM (green) and Mitotracker (red), respectively, in control or VAPB-silenced cells (details on the cell lines are in [94]). NSC34 cells were incubated for 30 min with TMRM and Deep Red-Mitotracker (both fromThermoFisher), each at 100 nM concentration, and then imaged alive, using the LSM800 confocal system equipped with an on-stage incubator (37 °C, 5% CO_2_) and a PlanApo 63 × 1.4 N.A. objective. The lack of bleedthrough between the red and far red signals was checked in singly labelled specimens. The images show single confocal sections, with the merged signal from the two channels. The prevalence of the green pseudocolour in the image of the control versus the silenced cells indicates a higher TMRM to Mitotracker ratio. Acquisition parameters and subsequent adjustment with Adobe Photoshop software were kept identical for the two illustrated images. Scale bar, 10 μM. (**B**) Quantification of TMRM to Mitotracker fluorescence in control and VAPB-silenced cells. The analysis was carried out on stacks of 5–7 sections of cells, imaged as in panel (**A**), keeping the illumination conditions so as to avoid any saturation of the signal. A mask of the Mitotracker signal was created using ImageJ software, and the integrated intensity of the Mitotracker and TMRM signals determined within the mask on each section, and then summed over the entire stack. The ratios of the summed integrated intensities for each individual stack (acquired in seven independent experiments), together with the medians and interquartile range are shown. * *p* = 0.025 by Student’s two-tailed *t*-test.

**Figure 3 cells-10-01865-f003:**
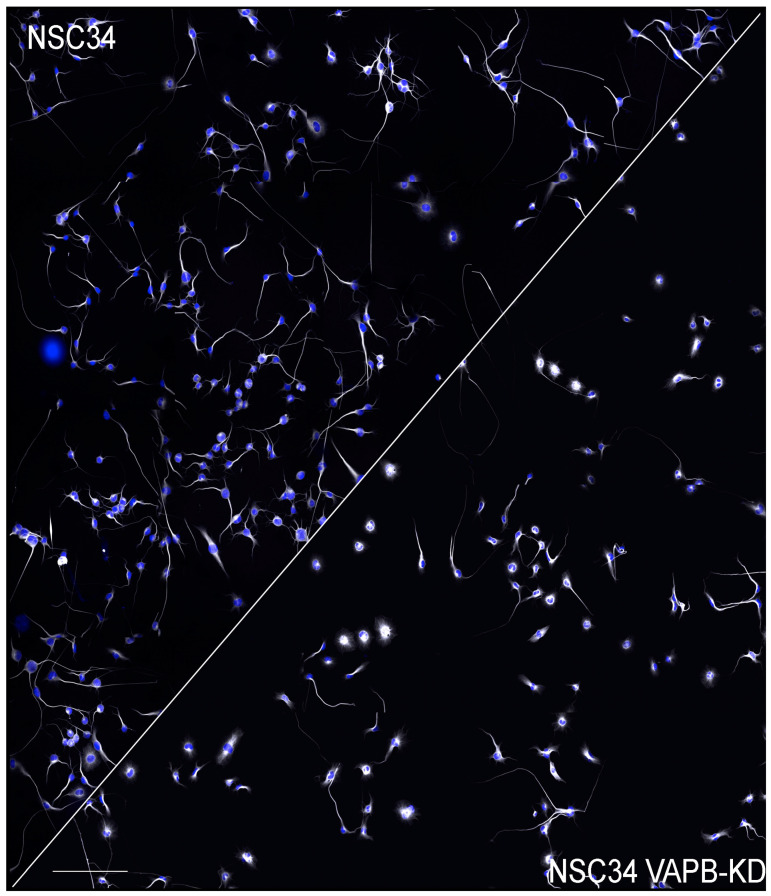
Delayed neurite elongation in NSC34 cells with downregulated VAPB. Low magnification fields of control (left) and *VAPB*-dsilenced NSC34 cells allowed to differentiate for 3 days. Nuclei are shown in blue and tubulin staining is in greyscale. The image of the control cells is reproduced from [94]. The reader is referred to that publication for details. Scale bar, 300 μm.

**Table 1 cells-10-01865-t001:** Interactions of the VAP proteins and their functional roles *.

	VAP Interacting Protein	Functions	References
**Interactions at Contact Sites**			
*ER-Golgi complex*	OSBP ^#^	Lipid Transport Protein (LTP) that regulates phosphatidylinositol-4-phosphate (PI4P) and cholesterol levels at Golgi membranes, by transferring cholesterol from the ER to the Golgi with back transfer of PI4P from the Golgi to the ER.	[45]
	CERT	LTP that transfers ceramide (precursor of glycosphingolipids and sphingomyelin) from the ER to the Golgi	[46,47]
	FAPP2	LTP that mediates glucosylceramide transfer to the trans Golgi. The lipid-transfer activity of FAPP2 is required for its role in membrane trafficking.	[39,48]
	FAPP1	In addition to interacting with VAP, it binds the ER phosphoinositide phosphatase SAC1, allowing it to hydrolyse PI4P in *trans* at the Golgi.	[49]
	NIR2	Transport of PI from the ER to the Golgi, where it is phosphorylated to generate PI4P.	[50]
*ER-Plasma* *Membrane (PM)*	NIR2/3	Transfer of phosphatidylinositol (PI) from the ER to the PM and delivery of phosphatidic acid from the PM to the ER. Maintenance of PM lipid composition and identity.	[51]
	Kv2.1 Potassium channels ^§^	Creation of dynamic membrane microdomains for potassium channels clustering at PM. VAP-Kv2 interaction facilitates recruitment to the PM of Nir2/3 PI transfer proteins, thus contributing to phosphoinositide homeostasis.	[40,52]
	ORP3 ^§^	Regulation of PI4P homeostasis and Ca^++^ dynamics by activating Protein Kinase C. Interacts with the small GTPase R-Ras, regulating cell adhesion, spreading and migration. Involved in the formation of membrane protrusions and in the regulation of actin cytoskeleton	[53,54]
*ER-Endosomes/Lysosomes*	StARD3, StARD3NL	Cholesterol sensing and regulation of endosome morphology, positioning and dynamics	[55]
	OSBP-related Protein ORP1L	Cholesterol sensing; regulates cholesterol egress from the endo-lysosomal system; negatively regulates dynein association with Late Endosomes (LEs)	[56,57]
	OSBP	PI4P transport from endosomes to the ER	[58]
	VPS 13C	Non vesicular lipid transfer	[59]
	Retromer SNX2 subunit	SNX2 tethers ER to endosomes through VAP at sites of actin- regulated budding	[58]
*ER-Mitochondria*	PTPIP51	Calcium homeostasis and regulation of mitochondrial energy metabolism; Autophagy	[60,61,62]
	VPS13 A and D ^§^	Non vesicular lipid transport through hydrophobic channel; Regulation of mitochondria size, shape and clearance	[42,59,63]
	α-synuclein	Overexpression of wild-type and familial Parkinson’s disease mutant α -synuclein disrupts the VAPB-PTPIP51 tethers and loosens ER–mitochondria associations	[64]
	MIGA-2 ^§^	Outer mitochondrial membrane protein that mediates a three-way contact between the ER, mitochondria and LDs. Coordination of mitochondrial metabolism with triglyceride production in the ER, facilitating lipid storage in LDs and promoting adipocyte differentiation	[65,66]
*ER-lipid droplets*	VPS13A VPS13C	Non vesicular lipid transfer	[59]
*ER-Peroxisomes*	VPS13D	Non vesicular lipid transfer. Regulation of peroxisomal biogenesis	[67]
	ACBD5	Regulation of peroxisome motility and growth	[68,69]
*ER-Isolation Membrane*	FIP200 ULK1 WIPI2	Interaction with these autophagy proteins modulates autophagosome biogenesis.	[70]
**Interactions within the ER**			
	**Protrudin**	VAP -protrudin interaction is required for protrudin’s function at Late Endosome (LE)–ER contacts. Protrudin transfers kinesin-1 from the ER to LEs thereby promoting microtubule-dependent translocation of LEs to the cell periphery and neurite elongation	[71,72]
	YIF1A	ER-Golgi trafficking protein regulated by VAP. The interaction is mediated by VAP’s TM domain and is important for both axon and dendrite extension	[73]
	SAC1	This phosphoinositide phosphatase hydrolyses PI4P to PI in the ER, thus maintaining a PI4P chemical gradient between the Golgi and the ER, which drives OSBP-mediated cholesterol/PI4P exchange at Golgi-ER contact sites. The interaction with VAPB recruits SAC1 to these contact sites.	[49,74]
	ATF6	The interaction with VAP may attenuate ATF6’s transcriptional activity, thus regulating ER stress.	[75]
	HCN Channels	Hyperpolarisation-activated cyclic nucleotide-gated channels that play a key role in the regulation of cardiac and neuronal pacemaker depolarisation. VAPB favours channel expression on the cell surface. The interaction is mediated by the TM domains.	[76]
**Nuclear Envelope**			
	Emerin	Inner nuclear membrane protein involved in nuclear envelope assembly. Loss of VAPB causes delocalisation of emerin to a cytoplasmic compartment.	[77,78]
	ELYS	Nucleporin required for nuclear pore assembly	[78,79]
**Other Interactions**			
	**CALCOCO1**	CALCOCO1 is an ER-phagy receptor	[80]
	Rab3 GTPase activating protein 1(Rab3GAP1)	Implicated in nuclear envelope formation	[81]
	Secernin 1 (SCRN1)	Modulation of Ca^++^ dynamics and synaptic vesicle cycling at presynaptic sites	[82]
	FAF1	Ubiquitin-binding adaptor for the AAA ATPase p97/VCP, involved in retrotraslocation of proteins from the ER to the cytosol in the ER-associated Degradation Pathway (ERAD)	[83]
	AKAP 220 and 110	Recruitment to the ER of PKA	[39]
**Viral Proteins**			
	Hepatitis C virus (HCV) NS5A and NS5B non-structural proteins	Interaction of these viral nonstructural proteins with the VAPs is required for viral replication. The VAPs contribute to anchoring, assembly and functioning of the viral replication machinery in close contact with the host cell ER membrane. In particular, recruitment of host cell VAP-interacting proteins (such as OSBP/PI4P/SAC1) results in the formation of membrane contact sites (MCS) between the ER and viral replication organelles (RO) and transfer of lipids (cholesterol) to the RO membrane (reviewed in [84,85])	[86,87,88,89]
	NS3-4A HCV non-structural protease complex		[90]
	Norovirus non-structural proteins NS1/2		[91]
	Aichi Virus non structural proteins		[92]
**Bacterial Proteins**			
	IncV Chlamydia inclusion membrane protein	IncV promotes the formation of membrane contact sites between the host ER and the pathogen-containing vacuole in a VAP-dependent manner.	[93]

* The table reports only the results obtained in mammalian cells and is limited to interactions whose functional significance is supported experimentally. ^#^ Proteins indicated in **red** interact with VAPs through a FFAT or FFAT-like motif, while proteins whose interactions are not mediated by the FFAT motif are indicated in **blue**. ^§^ Phospho-FFAT motif.

## Data Availability

Not applicable.

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
