# Peer review of "The Link between VAPB Loss of Function and Amyotrophic Lateral Sclerosis"

_cells, 2021, doi:10.3390/cells10081865_

Round 1
Reviewer 1 Report
The Manuscript is very interesting, as it performs a comprehensive bibliographic review on the many aspects of ALS8 pathological process, in different experimental models. It describes in detail the main physiological processes already found to be deregulated, such as ER – mitochondria crosstalk, autophagy and ER stress. A criticism, however, is that synaptic alterations caused by VAPB loss of function are only scantly mentioned. As a synaptopathy, the final result of ALS pathological alterations are synaptic loss and neurodegeneration, and this could be more explored. The relationship among the different markers of ALS8 pathology and motor neuron synapsis would also be important to be highlighted.
Author Response
Response to Reviewer #1.
- Although appreciating the interest of our manuscript, the reviewer noticed that "synaptic alterations caused by VAPB loss of function are only scantly mentioned". Given the importance of synaptic loss in ALS, he/she felt that more emphasis should have been placed on this aspect.
OUR REPLY: we fully agree that loss of neuromuscular junctions is a key aspect of ALS progression, however, not much information is available on this subject in VAPB loss-of-function models, the focus of our review. To address the concern of the reviewer, we now refer to synaptic alterations, in the cases where they have been analysed. Specifically: in section 3 (paragraph 4, p. 10 of revised manuscript), we now state that the P56S-VAPB knock-in mouse developed by Larroquette et al. developed both motor dysfunction and partial denervation of lower MNs; in section 4.2 (last paragraph, p. 15 of revised manuscript), we have added the expression "alterations of synaptic morphology" to the description of the work of Forrest et al. and Zhang et al. on invertebrate models; in section 4.4.3 (second paragraph, p. 21 of revised manuscript), we again refer to the "morphological alterations of the neuromuscular junction and partial muscle denervation" described by Larroquette et al., in the mouse P56S-VAPB knock-in model.
- The reviewer also felt that "the relationship among the different markers of ALS8 pathology and motor neuron synapses would also be important to be highlighted".
OUR REPLY: While we agree that the investigation of markers of ALS8 pathology is an extremely important subject, of paramount importance for the clinic, this subject is beyond the scope of our review, which specifically addresses the cellular-physiological consequences of VAPB loss-of-function in motor neurons. Nevertheless, we now mention the possibility that VAPB malfolding in peripheral blood mononuclear cells could represent a novel diagnostic marker for ALS, as suggested in the study of Cadoni et al. (section 5 - last paragraph, p. 23 of revised manuscript).
Reviewer 2 Report
The review titled “The Link between VAPB Loss of Function and Amyotrophic Lateral Sclerosis”, by Francesca Navone is well-organized and covers the area of research well. This manuscript well describes the results of various studies on the roles of VAPB in ALS disease. Here are my minor comments.
- There should be a Figure showing the effects of VAPB depletion in cellular and animal models. VAPB depletion has diverse effects on a wide range of areas, such as ER-mitochondrial contact, regulation of PI4P, nuclear envelope defects, delayed neurogenesis, HCN channels, UPR, and PQC. For each major topic, authors are required to include a Figure summarizing the molecules interacting with VAPB and the consequences of VAPB depletion.
- “RNAse Ire1” in line 619 should be corrected to “RNase IRE1”.
- The text of “the RNAse activity of XBP1” in line 622 should be corrected to “the RNase activity of IRE1”.
- “Ire1” in line 648 should be corrected to “IRE1”.
- Spliced Xbp1 mRNA encodes the transcription factor XBP1s. Author should distinguish between XBP1 and XBP1s in this manuscript.
- There are citation error on line 719 and 728.
Author Response
Response to Reviewer #2.
The reviewer appreciated out work, but has six minor comments for improvement.
- The reviewer feels that the article is under-illustrated. He/she states "For each major topic, the authors are required to include a Figure summarizing the molecules interacting with VAPB and the consequences of VAPB depletion."
OUR REPLY: We think that Figure 1C addresses the concern of the reviewer. Indeed the four boxes contained in the panel each illustrate the molecular interactions that are described in section 4 of the text. To more clearly connect this figure with the text, we have slightly modified the figure legend, and, importantly, at the beginning of each subsection of section four, we refer to the relevant box of the figure.
2 and 4: Replace Ire1 with IRE1.
OUR REPLY: This has been done (pp. 19-20 of the revised manuscript).
- Erroneous mix-up between XBP1 and IRE1.
OUR REPLY: This has been corrected (p. 19 of revised manuscript).
- Refer to the active product of spliced Xbp1 mRNA as XBP1s.
OUR REPLY: This has been done (pp. 19-22 of revised manuscript).
- Two citation errors
OUR REPLY: The unformatted references have now been correctly formatted.